



# Recession discharge from compartmentalized bedrock hillslopes

Clément Roques[1,2], David E. Rupp[3], Jean-Raynald de Dreuzy[1], Laurent Longuevergne[1], Elizabeth R. Jachens[4], Gordon Grant[5], Luc Aquilina[1] and John S. Selker[4]

[1]Univ Rennes 1, CNRS, Géosciences Rennes - UMR 6118, F-35000 Rennes, France.
[2]Centre for Hydrology and Geothermics (CHYN), Université de Neuchâtel, Neuchâtel, Switzerland.
[3]Oregon Climate Change Research Institute, College of Earth, Ocean, and Atmospheric Sciences, Oregon State University, Corvallis, OR, USA.
[4]Biological and Ecological Engineering Department, Oregon State University, Corvallis, OR, USA.
[5]Pacific Northwest Research Station, Forest Service, U.S. Department of Agriculture, Corvallis, Oregon, USA.

*Correspondence to*: Clément Roques, clement.roques@unine.ch

**Abstract.** We used numerical modelling to explore the role of vertical compartmentalization of hillslopes on groundwater flow and recession discharge. We found that when hydraulic properties are vertically compartmentalized, streamflow recession behaviour may strongly deviate from what is predicted by groundwater theory that considers the drainage of shallow reservoirs with homogeneous properties. We further identified the hillslope configurations for which the homogeneous theory derived from the Boussinesq solution approximately hold and conversely for which it fails. By comparing the modelled recession discharge and the groundwater table dynamics, we identified the critical hydrogeological conditions responsible for the emergence of strong deviations. The three main controls are i) the contribution of a deep aquifer connected to the stream, ii) the heterogeneity in hydraulic properties, and iii) the slope of the interface between shallow permeable compartment and deep bedrock one with lower hydraulic properties. Our results confirm that a physical interpretation of the recession discharge exponent $b$ from the classical equation $-dQ/dt = aQ^b$, and its temporal progression, are information that can only be interpreted with knowledge on structural configuration and heterogeneity of the aquifer.

## 1 Introduction

Recent streamflow recession analyses have provided new insight into the spatial and temporal variability of hydrological behaviour across the globe (Fan, 2015; Roques et al., 2021; Schmidt et al., 2020; Tashie et al., 2020). These analyses have confirmed that the relationship between groundwater storage and stream discharge (hereafter called storage-discharge functions) may be highly nonlinear and deviate from what would be predicted from groundwater theory under the assumption of homogeneous subsurface properties. They have also shown that such non-linearities in storage-discharge functions may be variable across different recharge events, suggesting strong dependencies to initial saturation conditions (Jachens et al., 2020; Tashie et al., 2020). In the current context where hydrological systems are increasingly impacted by more frequent and intense periods of droughts, pushing aquifers and streams to new extremes (Famiglietti, 2014; Gudmundsson et al., 2021), it is urgent





that we increase understanding of the main controlling factors responsible for such behavior (Blöschl et al., 2019; Fan et al., 2019).

Groundwater theory assuming homogeneous subsurface properties shows that the rate of change in discharge $-dQ/dt$ of the
falling limb of the hydrograph may scale as a power law of the actual discharge $Q$:

$$-\frac{dQ}{dt} = aQ^b \qquad \text{Eq. 1}$$

The recession constants *a* and *b* arise from the geometric and hydraulic properties of the aquifer system. They are also highly sensitive to the predominant diffusive or kinematic flow conditions (Rupp and Selker, 2006). *b* is specifically defined as the constant of non-linearity in the storage-discharge function (Kirchner, 2009). Large values of *b* are characteristic of resilient streams, whose low flows decrease slowly with time (Berghuijs et al., 2016; Jachens et al., 2020; Kirchner, 2009). Models
derived from solutions to the 1D Boussinesq equation assuming homogeneous and horizontal aquifers predict values from 1 to 1.5 under low-flow conditions (Brutsaert and Nieber, 1977): *b* ranges from *b*=1 for confined and thick unconfined aquifers (i.e. much thicker than the depth of the penetrating stream channel), whose saturated thickness only marginally changes, to *b*=1.5 for a relatively thin unconfined aquifer (Brutsaert and Nieber, 1977; Troch et al., 2013; Wittenberg, 1999). When *b*=1, the aquifer drainage is exponential with a characteristic timescale $\tau = 1/a$. When *b*=1.5, the decrease of transmissivity with
head slows down the aquifer drainage. There is currently a dearth in the definition of contexts where such simplified effective representations are likely to succeed or fail in accurately describing groundwater storage-discharge dynamics.

Indeed, observed values of *b* for low-flow conditions may exceed the theoretical values previously described, questioning the use of homogeneous groundwater flow theories (Jachens et al., 2020; Roques et al., 2017, 2021; Tashie et al., 2020). Previous
authors have explored the processes that might be responsible. They include the effect of spatial and vertical distribution of flow paths (Clark et al., 2009; Harman et al., 2009; Jachens et al., 2020; Rupp and Selker, 2005), the influence of planform shape of elementary area on flow paths (Paniconi et al., 2003), the spatial variability of recharge and evapotranspiration (Tashie et al., 2020), and the storage from the unsaturated zone (Luo et al., 2018). Quantifying the relative contribution of all those factors in controlling the storage-discharge non-linearity remains a major challenge in hydrology.


The storage-discharge function becomes non-linear in catchments where geological heterogeneities affect the partitioning of groundwater flow. Heterogeneity may arise from the presence of different lithologies in the catchment with contrasted permeability. Another source of heterogeneity, which has been described in reference studies, is the general decrease of permeability with depth (Anderson, 2015; Boutt et al., 2010; Clair et al., 2015; Guihéneuf et al., 2014; Hencher et al., 2011;
Welch and Allen, 2014). All conceptual models agree that: 1) hydraulic conductivity decreases with depth in response to increasing confining stress and illuviation. Permeability data inferred from borehole tests show an  exponential trend with depth that have been quantified in several studies around the world (Achtziger-Zupančič et al., 2017; Ingebritsen and Manning,





1999; Saar and Manga, 2004), and 2) hydraulic properties may be enhanced in the shallower part (up to a depth of 100 meters) due to supergene weathering processes and the presence of elevated joint densities in response to unloading and topographic

stresses (Dewandel et al., 2006; Harman and Cosans, 2019; Moon et al., 2017; Rempe and Dietrich, 2014; Riebe et al., 2017). Besides such evidence of the vertical compartmentalization of hillslopes, efforts aimed at modelling groundwater storage variations and streamflow discharge often simplify this heterogeneity by considering a homogeneous aquifer with effective hydraulic properties. Here we aim at identifying in which context this simplification may hold and conversely in which ones it could be a source of large uncertainty and error when estimating storage-discharge functions of hillslopes.


Storage-discharge functions with values of $b$ larger than 1.5 during baseflow suggest that the resistance to drainage can be even stronger than the one induced by the decrease of transmissivity with the hydraulic head. The decrease of hydraulic conductivity with depth and vertical compartmentalization plays a significant role in the emergence of anomalous $b$ values. In this perspective, Rupp and Selker derived analytical solutions for the case of horizontal (2005) and sloping (2006) aquifers

with a power law decrease of hydraulic conductivity with depth such that $K \cong z^n$, where $z$ is the elevation and $n$ is the power-law exponent ($n > 0$). The authors showed that $b$ is determined by $n$ according to $b = \frac{2n+3}{n+2}$ for a horizontal aquifer, such that $b$ is an increasing function of $n$ with a lower limit of 1.5 for $n$ near zero to an upper limit of 2 for large values of $n$. As expected, the stratification of the hydraulic conductivity increases $b$ and slows down the drainage. In this approach, groundwater flows are constrained to the upper-most material, implicitly supposing that there is a shallow impermeable boundary parallel to

topography. Although this may be a reasonable assumption in some cases, we hypothesize in this paper that, in other contexts, deeper aquifers may also significantly contribute to stream discharge (Figure 1a.). To test our hypothesis, we determine the recession discharge behaviour arising from the presence within the hillslope of a shallower compartment overlaying on a deeper one with contrasted hydraulic conductivities. This structure typically represents a permeable near-surface regolith aquifer underlain by a deeper less permeable bedrock compartment (Figure 1a). Such structures are expected to induce the two

following effects. First, the aquifer discharge is expected to slow down progressively from the fast drainage of the permeable near-surface layer with low characteristic timescales (high value of $a$ as pictured in blue in Figure 1b and c) to the drainage of the deeper less permeable compartment with consequently longer drainage timescale (lower value of $a$ as pictured in red in Figure 1b and c). Second, the sloping interface between the two compartments conditions the shape of the water table profile and progressively modifies the partitioning of flows on either side of the interface. The transition between the trends given by

each of the compartments is likely to cover a wide range of times and behaviours. For a simpler system of two parallel reservoirs (Figure 1b-d), Gao et al (2017) have shown a broad variety of transitioning flows and found that $b$ may be written as a function of the ratio of the fast flow to the total flow and of the hydraulic conductivity ratio between the two compartments (Eq. 19 of their paper and in Appendix A.1). We will use the two-linear reservoir theory as a reference to describe our results and explore in which context such simplification might hold to describe storage-discharge functions of vertically

compartmented hillslopes.


**Figure 1: a. Conceptual model of a compartmentalized hillslope with groundwater flow distributed between fast flow within the shallow regolith/fractured aquifer and slow flow in the underlain bedrock; b. Theoretical streamflow behaviour considering two linear reservoirs (b=1) in parallel with fast (blue line, τ= 10 d) and slow (red line, τ= 100 d) drainage timescales; c. resulting recession plot -dQ/dt = f(Q); and, d. the evolution of b as a function of discharge identifying high values during the transitional flow regimes up to 3 in this case.**






## 2 Methodology

We investigate the time dependency of streamflow recession behaviours during the drainage of hillslopes where both shallow
and deep compartments are involved (Figure 1a.). We specifically provide an exhaustive analysis of the respective controls of
both geometrical and hydraulic properties in controlling storage-discharge non-linearity in our two-compartment
representation of the subsurface.

### 2.1 Model geometry and structure

We consider a 2-dimensional representation of the subsurface (Figure 2) which consists of a lower compartment with a
horizontal impermeable base and a length $L = 100$ m from the stream location to the groundwater divide. The elevation $z = 0$
is set at the river elevation located on the left boundary. The bottom of the model is at $-\zeta$ at $x = 0$. We define an upper
compartment of constant thickness $D$, leading to a total thickness $(D + \zeta)$ at $x = 0$ that is equal to $L$, and a slope angle $\beta =
e/L$, where $e$ is the elevation of the interface at $x = L$.

In the lower compartment, the saturated hydraulic conductivity $K_L$ and the porosity $\theta_L$ decrease exponentially with depth
following the classically used relationship (e.g. in Cardenas and Jiang (2010)):

$$K_L\ (x,z) = K_{L(x=0,z=0)} \exp\left(-\frac{1}{\alpha}\left(z_s\ (x) - z\right)\right) \qquad \text{Eq. 2}$$

and

$$\theta_L\ (x,z) = \theta_{L(x=0,z=0)} \exp\left(-\frac{1}{\alpha\eta}\left(z_s\ (x) - z\right)\right) \qquad \text{Eq. 3}$$

where $z_s = \beta x$ is the elevation of the interface between the two compartments and $\alpha$ is the characteristic depth over which
hydraulic properties decrease. Initial values of $K_{L(x=0,z=0)}$ and $\theta_{L(x=0,z=0)}$ were taken arbitrarily at $5\ 10^{-6}\ m/s$ and 0.3
respectively. Note that we found that changing the parameter $\alpha$ does not significantly impact our results (see Appendix A. 2).
$\eta$ is a coefficient related to the medium structure that we choose equal to 2 as commonly reported in the literature (Bernabé et
al., 2003; Cardenas and Jiang, 2010). The upper layer has a homogeneous isotropic saturated hydraulic conductivity and a
homogeneous porosity. We explore contrasts in hydraulic conductivities between both compartments according to the ratio $r_K$
as $K_U = r_K\ K_{L(x=0,z=0)}$. The porosity of the upper compartment and the interface remain constant across simulations with $\theta_U =
\theta_{L(x=0,z=0)}$.





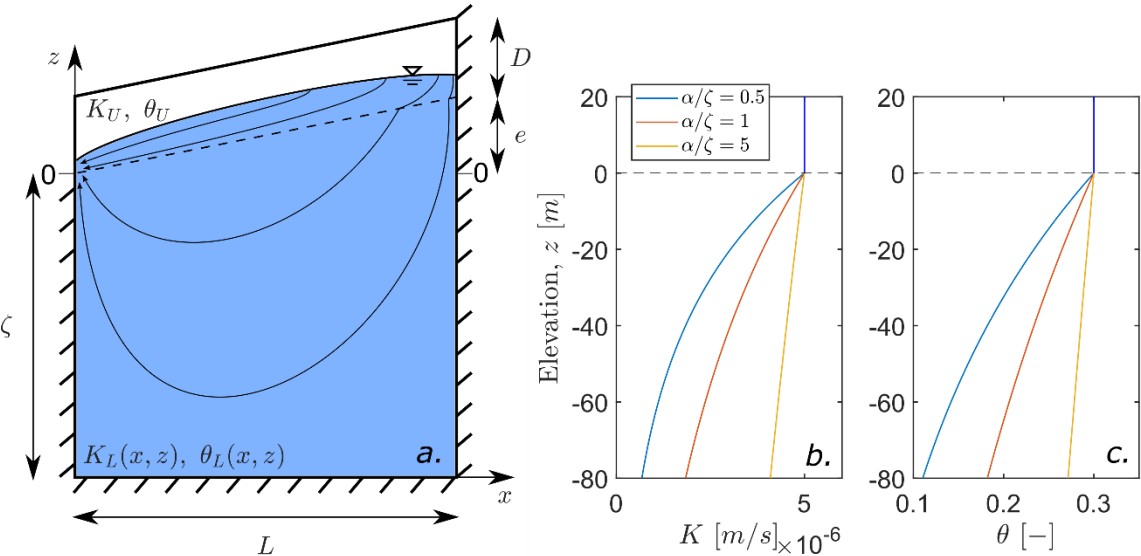

**Figure 2: a. 2D model cross-section with the main geometrical parameters identified along with model boundaries and groundwater flow lines, b. and c. are vertical profiles of $K$ and $\theta$ respectively for different $\alpha/\zeta$ values investigated taken at $x = 0$.**

The geometrical parameters $\beta$ and $D$ are varied in order to explore different hillslope configurations, with $D/L = [0.1, 0.2, 0.5]$ and $\beta = e / L = [0.0, 0.1, 0.2, 0.5]$. The exponential decrease in hydraulic properties of the lower compartment is changed by varying $\alpha/\zeta = [0.5, 1, 5]$ in Eq. 2 and Eq. 3. Finally, the contrast in hydraulic conductivities is explored over two orders of magnitude with $r_K = [1, 4, 16, 64, 256]$.

**2.2 Groundwater flow simulation**

We performed Computational Fluid Dynamics (CFD) simulations in the 2-dimensional (2D) hillslope. The flow for variably saturated porous media with incompressible fluid is modelled by solving the Richards' equations with a finite element approach. We set the vertical uphill boundary to no-flow. The vertical downhill boundary is divided into a no-flow boundary below the river elevation and a fixed-head boundary on the top layer. The fixed-head boundary allows groundwater to discharge into the stream at a rate that is driven by the head difference between the aquifer and the river and a conductance term. For

simplicity, we set the head of the river to 0 m, meaning that no river stage variation is allowed in our model. We set the conductance between the aquifer and stream equal to the hydraulic conductivity of the upper most aquifer $K_U$ to avoid any additional resistance at the boundary. We assume that the rock compressibility is negligible - by setting it equal to the water compressibility - implying that the storage coefficient only depends on the effective porosity.

The mesh is composed of tetrahedral elements. We imposed a finer mesh in the upper most part of the model where the fastest changes in water table elevation and saturation occur. The initial conditions are of a fully saturated aquifer which then drains towards an asymptotically flat-water table. We excluded the initial times of the simulations, until the water table progressed





toward a free water table profile from the river to the uphill no-flow boundary. The case of a horizontal homogeneous shallow aquifer, i.e. without the deeper compartment, was checked against the analytical solution from Boussinesq (Boussinesq, 1904)

to ensure accurate convergence, i.e. considering Eq. 1 for discharge per unit of river length $Q$, with $a$ and $b$ defined as:

$$a = \frac{4.804 K^{1/2}}{\theta (2L)^{3/2}} ; b = 1.5 \qquad \text{Eq. 4}$$

## 3 Results

### 3.1 Effect of the deep compartment in the case of a flat aquifer ($\beta = 0$)

First, we review the numerical results for a horizontal homogeneous shallow aquifer overlying a deeper, less permeable, compartment, *i.e.* the case where $\beta = e/L = 0$. In order to identify the relative influence of the deep aquifer on the recession

behaviour for the different values of $r_K$, we compare in Figure 3 our results with predictions obtained from the classical analytical solution Eq. 1 with recession constants from Eq. 4 which consider only the drainage of a horizontal shallow aquifer with homogeneous hydraulic properties (Boussinesq, 1904). For a review of the different analytical solutions available, the reader is referred to Rupp and Selker (2006) and Dewandel et al. (2003).

The $-dQ/dt$ vs $Q$ recession curves shift upward as $r_K$ increases, *i.e.* increasing rate of change in discharge for the same discharge value as $r_K$ increases, which corresponds to an increase in the value of the recession constant $a$ in Eq. 1. This is expected from Eq. 4, with $a$ being directly proportional to $\sqrt{K}$. We found that the analytical solution provides accurate results for high values of $r_K$ ($> 16$), while deviations are seen when $r_K$ is smaller than 16. At the late times, the analytical solution significantly underestimates $-dQ/dt$ with $b$ tending to lower values as recession progresses. For higher values of $r_K$, the

source of discharge is mainly the upper compartment which satisfies the assumption of only considering the shallower flat aquifer. However, for lower values of $r_K$, flow tends to be distributed over the full depth of the aquifer system, leading to higher $-dQ/dt$ as recession advances than the ones predicted by the analytical solution.

We expect that for low values of $r_K$, $b$ will take values close to 1, similar to the case of a thick aquifer with only marginal

transmissivity variations with head. In contrast, for a high ratio $r_K$, the change in transmissivity due to the fast drainage of the upper, more permeable, compartment satisfies the assumption of the Boussinesq solution for a relatively thin unconfined aquifer over a large range in $Q$. This comparison allows us to identify the influence of the deeper compartment in shaping the recession hydrograph with respect to $r_K$.



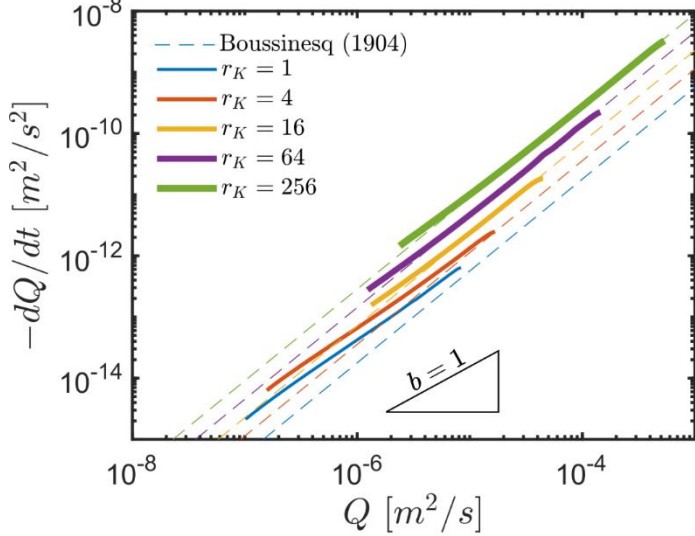


**Figure 3: Recession plots displaying $-dQ/dt$ vs $Q$ in a log-log space obtained for the horizontal interface, i.e. $\beta = 0$, and different hydraulic conductivity ratios $r_K$. The dashed lines show $-dQ/dt$ vs $Q$ predicted with the analytical solution for a flat and homogeneous aquifer without the deep compartment ($b = 1.5$; Boussinesq 1904). The slope for $b = 1$, characteristic of the drainage of a linear reservoir, has been drawn for reference.**

**3.2 Effect of an inclined interface ($\beta>0$)**

The variety of recession behaviours obtained by changing both $r_K$ and $\beta$ is displayed in Figure 4. Considering cases where $\beta > 0$ and by increasing $r_K$, the recessions transition from higher values of $-dQ/dt$ toward lower ones. With time, the water table progressively lowers below the interface inducing a sharp reduction in the effective hydraulic conductivity that translates to larger values of $b$. Finally, recession progressively converges to a value of $a$ that is characteristic of the geometrical and

hydraulic properties of the deeper compartment, *i.e.* the blue lines at late times for $r_K = 1$. The change of local $b$ values as a function of $Q$ for each hillslope configuration $b_i$ are displayed in Figure 5. It is computed as:

$$b_i = \log\left(\left(-\frac{dQ}{dt}\right)_{i+1} - \left(-\frac{dQ}{dt}\right)_i\right)/\log(Q_{i+1} - Q_i)$$

Eq. 5

where $i$ and $i + 1$ are indexes of the discharge range considered $[Q_{min} : Q_{max}]$.



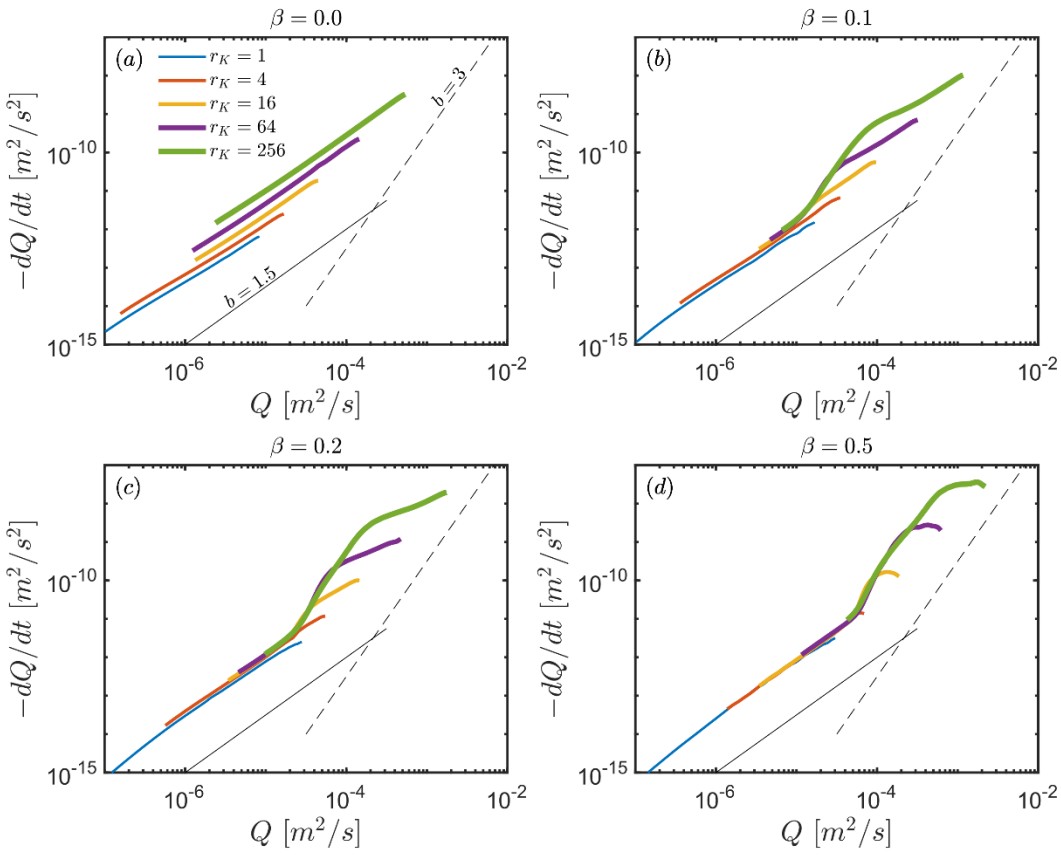

**Figure 4: Recession plots displaying $-dQ/dt$ vs $Q$ in a log-log space obtained for different hydraulic conductivity ratios $r_K$ and**
**different slope angles of the interface between shallow and deep compartments $\beta$. Lines of $b = 3$ and $b = 1.5$ are drawn for**
**references.**





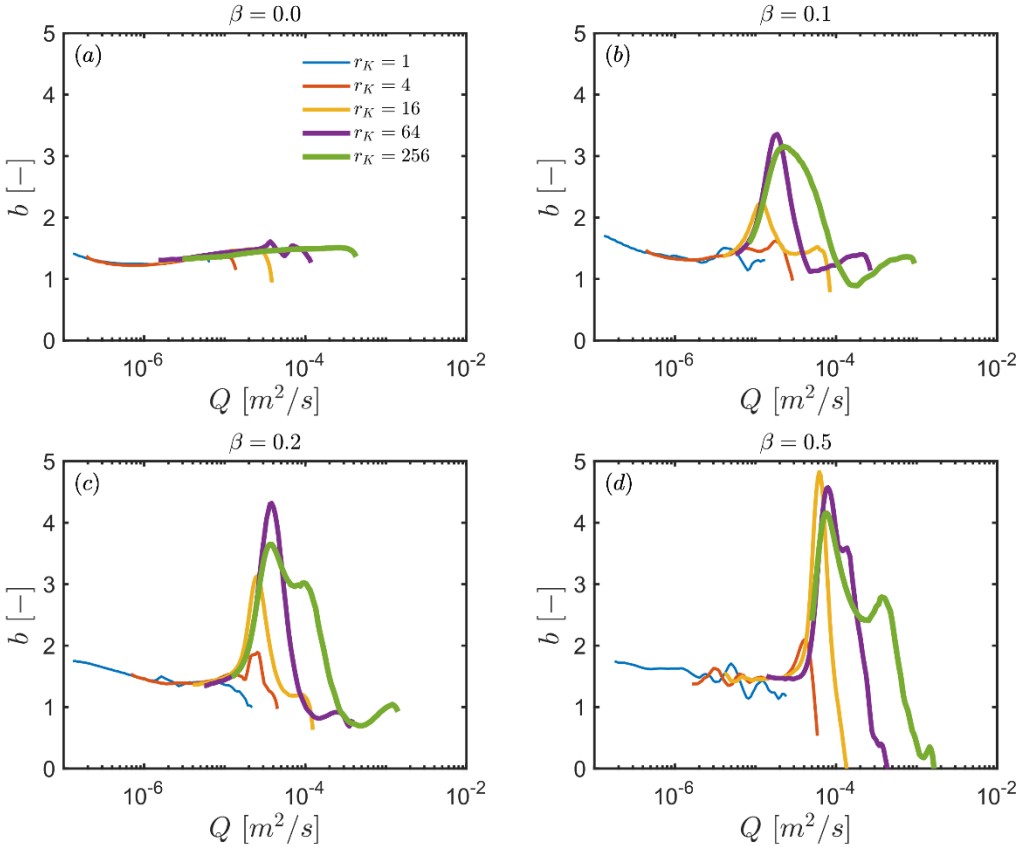

**Figure 5: Evolution of local b values as a function of discharge $Q$ for the range of hydraulic conductivity ratios $r_K$ and different slope angles of the interface between shallow and deep compartments $\beta$.**

Overall, sharp differences in recession behaviours are evident across all hillslope configurations. At late times, i.e. low discharge $Q$, all recessions show increasing $b$ towards, and ultimately above, 1.5 when flow is mostly controlled by the lower compartment. Theoretically, the drainage of a deep homogenous aquifer would lead to $b = 1$, i.e. negligible change in transmissivity during recession, whereas in our simulations, the exponential decrease in hydraulic conductivity with depth results in $b$ values $> 1$ at low $Q$. This latter result agrees qualitatively with the predictions made by Rupp and Selker (2005)

showing how a permeability that decreases smoothly with depth increases $b$ above the homogeneous case, up to a value of $b$ = 2. However, our configuration with a discontinuity of hydraulic conductivity, also shows how $b$ may not be constant during late times.

As expected from the solution considering the combined discharge from two linear reservoirs (Figure 1, Gao et al. 2017), the

maximum values of $b$ is strongly influenced by the contrast in hydraulic properties $r_K$ between the two overlapping compartments. We explore the impact of hillslope structural configurations on recession discharge by comparing our simulated





$b$ with the ones predicted from the two-linear reservoir theory. We derived the equation for the maximum value of $b$ from the equation provided by Gao et al. (2017) used to predict the evolution of $b$ as a function of discharge (see Appendix A1). We found $\max(b)$ to be only a function of $r_K$ between the two linear reservoirs:

$$\max(b) = \frac{(r_K + 1)^2}{4r_K} \qquad \text{Eq. 6}$$

We compare in Figure 6 the progression of $\max(b)$ as a function of $r_K$ and for different $\beta$ from our simulations against the prediction from Eq. 6. Our two-compartment model shows a non-monotonic behaviour. For $r_K \leq 4$, $\max(b)$ remains close to the case of the horizontal interface. However, for $r_K > 4$, $\max(b)$ significantly increases before decreasing back to lower values at high $r_K$. For example, in the case of $\beta = 0.5$, $\max(b)$ increases toward values close to 5 at $r_K = 16$ before decreasing back to values close to 4 at $r_K = 256$. This behaviour results from the different flow regimes involved.


Eq. 6 fails in predicting the variety of behaviours arising from the different subsurface configurations investigated here. While it might be useful in the case of lateral heterogeneity (Gao et al., 2017; Harman et al., 2009), the parallel reservoir paradigm is not able to predict the recession dynamics involved in the case of a vertically compartmentalized hillslope.

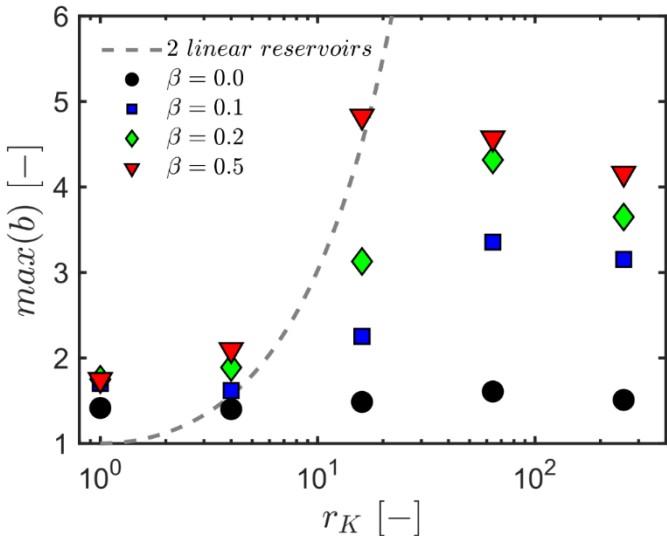

**Figure 6: Maximum values of b as a function of $r_K$. The different slope angles of the interface $\beta$ is pictured by different symbols. The grey dashed line represents which predict the evolution of maximum values of b as a function of $r_K$ considering the drainage of two-linear reservoirs in parallel (Gao et al., 2017).**

**3.3 Impact of transient flow regimes on $b$ values with steep interfaces ($\beta \gg 0$): transition from gravity- to diffusion-controlled regimes**

To illustrate the link between the transient pattern of the water table – and consequently the flow field – and its impact on the evolution of $b$, we show in Figure 7 the progression of the water table for two hillslopes under $\beta = 0.5$ with $r_K = 1$ (Figure 7a) and $r_K = 64$ (Figure 7b). We also plot in Figure 7c the progression of the average head gradient $\nabla H$ computed across the





hillslope, normalized by the slope of the interface $\beta$, and, in Figure 7d, the change in saturation area in the upper compartment, $\nu_u$, with respect to the total saturation area, $\nu$, as a function of normalized discharge rates $Q/Q_0$ for the same hillslope
configurations.

The water table profile for hillslope with $r_K = 1$ maintains a shape typical of diffusion-controlled drainage. This is confirmed by the linear decrease of the head gradient relative to discharge for $r_K = 1$ as expected from the Boussinesq equation (Figure 7c provided in linear scales in Appendix A3) and the progressive desaturation of the upper compartment (Figure 7d). In the
case of $r_K = 64$, the early stage of the recession follows almost constant $-dQ/dt$, with $b$ tending toward values close to 0 (even negative at the beginning of the recession) before increasing during a transition period (Figure 5d). This first regime where $b$ is close to 0 or negative suggests that drainage is controlled by a kinematic wave regime (Beven, 1982; Rupp and Selker, 2006) where the hydraulic head gradient mostly follows the slope interface, i.e., $\nabla H / \beta \approx 1$ (Figure 7c) and a constant or increasing rate of change in discharge is involved. This behaviour is visible in Figure 7b where the water table for $r_K = 64$
is mostly parallel to, and above, the interface near the start of the recession ($Q/Q_0 \approx 1$). As the recession advances, the water table drops rapidly below the interface (Figure 7b) with low saturation of the upper compartment $\nu_u$ (Figure 7d), whereby $\nabla H/\beta$ starts to decrease and discharge rates transition toward a diffusion-controlled regime. During the transition period the hydraulic gradient remains steep and the peak in $b$ values occurs. Finally, at the late times, the water table progressively disconnects from the interface and $b$ values decrease back toward 1.5 as expected from the diffusion-controlled regimes under
homogeneous conditions.



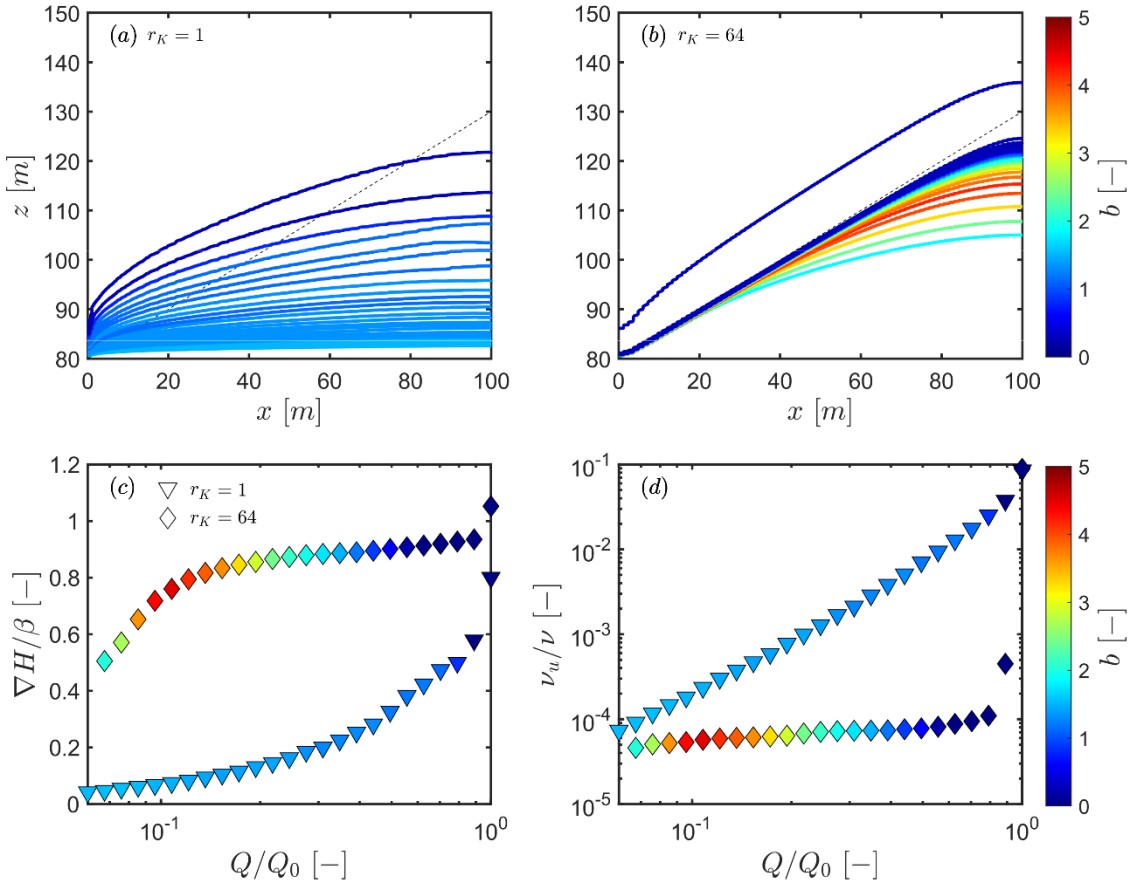

**Figure 7:** The upper most plots (a) and (b), represent the groundwater table profiles taken at different $Q/Q_0$ during the recession considering a) $\beta = 0.5$, $r_K = 1$ and b) $\beta = 0.5$, $r_K = 64$. The actual values are $Q/Q_0$ correspond to the ones used in the lower most plots (c) and (d). The interface is represented by the dashed black line in the background. Water tables are pictured as continuous line with colours scaling with the corresponding b values. The lower most plots show in (c) the progression of the average head gradient $\nabla H$, normalized by the slope of the interface $\beta$, and in (d) the change in saturation area of the upper compartment $\nu_u$ with respect to the total saturation area $\nu$, as a function of $Q/Q_0$ for the same hillslope configurations as in (a) and (b). Triangle shape symbols are for the case where $r_K = 1$ while diamond ones are for $r_K = 64$. The colours of the symbols show the magnitude of $b$ during recession.

## 4 Discussion

**In which geomorphological context might vertical compartmentalization impact recession discharge?** Heterogeneity is ubiquitous in the subsurface, inherited from the interplay between tectonic, erosion and weathering processes. St-Clair et al. (2015) presented a high-resolution geophysical imaging of bedrock in three US Critical Zone Observatories. They revealed a complex internal structure of hillslopes with clear differentiation between shallow permeable compartments overlying a deep and less permeable bedrock (Clair et al., 2015). There are multiple factors that may explain the presence of such compartmentalization. Several authors have revisited the mechanical processes that may be responsible for such



compartmentalization (Clair et al., 2015; Martel, 2006; Moon et al., 2017) and argued that near-surface tectonic and topographic stresses could be the main drivers for the development of permeable fractures organized sub-parallel to the topographic surface, also called exfoliation joints. Similarly, in volcanic regions, basalt formations have been described as highly compartmentalized due to the presence of high densities of fractures formed during cooling of the lava flow. Manga (1996), followed by several other works (Jefferson et al., 2006; Tague et al., 2008), have discussed the role of erosion and weathering in setting shorter drainage characteristic timescales in the Oregonian volcanic Cascades range. It has also been shown that, when hillslopes are under critical stress conditions, the permeability of the shallower compartment may be enhanced by gravitational movements which favour the development of rock damage and fracture density (Vallet et al., 2015; Wolter et al., 2020). Consequently, increased permeability and storage capacities in the shallow subsurface is a common characteristic in bedrock regions. Characterizing the depth and connectivity of such shallow compartments, set during the evolution of the landscape and overprinted by current stress and hydrogeological conditions, remains a main challenge (Dewandel et al., 2006; Guihéneuf et al., 2014; Litwin et al., 2021; Rempe and Dietrich, 2014; Riebe et al., 2017; Worthington et al., 2016).

While geophysics reveals the presence of such complexity with unprecedented details (Callahan et al., 2020; Clair et al., 2015; Flinchum et al., 2018; Leone et al., 2020), a challenge resides in attributing effective hydraulic properties to each compartment. This is classically achieved through borehole testing which provides precious local estimates. As examples, we could cite the studies performed in India (Dewandel et al., 2006; Guihéneuf et al., 2014; Maréchal et al., 2004), and the Armoricain massif (Ayraud et al., 2008; Le Borgne et al., 2006; Roques et al., 2014) that have revealed contrast in hydraulic conductivities distributed over 2-3 orders of magnitudes between shallow fractured/weathered aquifer and deeper "fresh" bedrock. Yet the question remains in the representativity of these local estimates in the description of hillslope to catchment-scale flow and transport processes.

**To what extent does vertical heterogeneity impact the discharge behaviour of hillslopes?** Despite our simplification of the actual heterogeneity of hillslopes, we reveal a strong diversity of impacts on recession behaviour across the different hillslope configurations investigated here. We found that, when compartmentalization is accounted for in the spatial distribution of hydraulic properties, recession behaviour strongly deviates from what is predicted by groundwater theory that considers the drainage of shallow reservoirs with homogeneous properties. Results specifically demonstrate that the critical parameters in the emergence of anomalous behaviour are *i)* the presence of a deep bedrock aquifer connected to the stream network, *ii)* the contrast in hydraulic properties between shallow and deep compartments, and *iii)* the slope of the interface (Figure 1). Other parameters such as the thickness of the shallow aquifer ($D$) and the decay exponent in hydraulic conductivity of the deeper compartment (parameter $\alpha$ in Eqs. 2 and 3) have less influence on the variability of recession exponent $b$ obtained in our simulations (Appendix A. 2). Noteworthy, our exploration of the decay exponent in hydraulic conductivity remains





restricted to a few cases of an exponential function. Other functions may deliver different results which would deserve a
dedicated study.

In the case of hillslopes with a horizontal interface between shallow and deep compartments, we reveal that there exists a
threshold in hydraulic conductivity ratio $r_K$ above which the assumption of only considering the drainage of a homogeneous

shallow aquifer derived from the Boussinesq solution seems to hold. In our simulation scheme, the assumption is satisfied for
$r_K \geq 16$ where lateral flow in the shallower compartment prevails, *i.e.* with only limited contribution from the deep bedrock
to the discharge dynamics. However, for lower values of $r_K$, the continuity in hydraulic properties between shallow and deep
compartments favour deeper flow partitioning. In this case, the recession behaviour deviates from the prediction of the
Boussineq equation. The values of $b$ tend toward values close to 1, typical of the drainage of thick aquifers where the change

in transmissivity involved by the desaturation during recession is negligible. At later times, $b$ values were found to increase
again controlled by the decay in hydraulic conductivity. This shows the intrinsic variability of recession behaviour, even in a
relatively simple configuration, which challenges the use of homogeneous groundwater theories.

Considering high slope interface, $\beta = 0.5$, and a strong contrast in hydraulic conductivities, $r_K = 64$, sharp changes in flow

regimes are involved when the water table transitions from the shallower compartment toward the deeper one with three
separated phases (Figure 7c-d). In the first phase, during high flow regimes when $Q/Q_0 > 0.8$ (i.e. the two first points from
the left of Figure 7c-d), the saturated volume drops by a factor of 4 with a minimal change in discharge rates (Figure 7d). The
upper zone drains quickly because of the combined effect of its higher conductivity and of the inclined interface. The slope
has a strong control on the head gradient, causing high discharge rates, even achieving a kinematic flow regime. Flows occur

predominantly in the upper compartment. In the second phase, discharge rates decrease sharply while transitioning to the lower
compartment at an almost constant head gradient. The sharp increase of the value of $b$ can be explained in the following way.
As $b = d(log(-dQ/dt))/d(logQ)$, and $Q \sim HK\nabla H$, for $b$ to be large, $K$ must be decreasing rapidly with the otherwise
steady $H\nabla H$ values. In the third phase, the head gradient decreases again with a flow mostly occurring in the lower
compartment. The value of $b$ decreases quickly with $Q$. The water table becomes parabolic uphill and remains influenced by

the proximity of the interface close to the river (Figure 7b) explaining the still significantly larger values of $b$ than in the flat
interface case (Figure 7a). The presence and geometry of the interface between high and low permeable compartments, and
the decreasing trend of permeability with depth are keys in conditioning the emergence of complex groundwater flow processes
and recession discharge behaviours. It is specifically true in this hillslope configuration for which the river channel fully
penetrates the upper compartment. This can be considered as common case for landscapes under sufficient erosion rates to

carve the weaker compartment, *i.e.* concerned by higher fracture density or higher porosity, until a more resistant fresh bedrock
is reached. The case where high erosion rates may lead to the formation of gorges and canyons in the fresh bedrock may result
in different behaviours. The impact of such specific configurations on recession behaviour remains to be explored.





**Knowledge regarding the structural configuration and heterogeneity of the aquifer is crucial when interpreting discharge recession behaviours.** Our results confirm that the exponent $b$ and its temporal progression are information that
can only be interpreted with some amount of local geological knowledge. One of the main challenges in hydrology is to identify the relative contribution among climatic, geomorphological, geological and land use factors in controlling spatial and temporal variabilities of discharge recession behaviour (Blöschl et al., 2019; Gnann et al., 2021). Our results suggest that different recession behaviour is expected depending on the structural configurations, the initial saturation condition and location of the water table prior recession. These are likely to be major factors in explaining the high variabilities in recession constants that
are observed in some catchments by recent works (Roques et al., 2017, 2021; Tashie et al., 2020) as it is for transit times and solute transport processes (Ameli et al., 2016; Berghuijs and Kirchner, 2017). We reveal that the temporal progression of head gradient and the relative proportion of saturation volume in the upper compartment with respect to the deeper one are key indicators to predict recession discharge behaviour. These indicators can be assessed through borehole logging and monitoring. The vertical compartmentalization is classically described through detailed core analysis and geophysical investigation (in
hole and surface). The interface can also be approximated through the maximum depth of the boreholes in a region. Indeed, it is common that the drilling of a borehole is stopped when a less productive zone is reached, i.e. when airlift discharges stop increases with depth. While such methods will inform about the geometry and heterogeneity of the hillslope, the pressure monitoring of a network of boreholes distributed along the hillslope will help identify the different flow regimes involved during recession. It can specifically provide estimates in both the progression of head gradients and saturation distribution
between upper and deeper compartments that, as we have shown here, are critical parameters to consider when interpreting streamflow recession dynamics.

## 5 Conclusion

While it is well accepted from structural and geophysical investigations that hillslopes settled in bedrocks are compartmentalized, its consideration is often overlooked in hydrological models used to predict the evolution of groundwater
storage and stream discharge dynamics (Blöschl et al., 2019; Clark et al., 2017; Gnann et al., 2021; Stephens et al., 2021). It is urgent to further develop knowledge on the role of such vertical compartmentalization and its representation at the catchment scale to improve prediction in stream baseflow regimes. Our results allow us to identify that both slope interface between deep bedrock and fractured/weathered shallow compartment and their contrast in hydraulic properties are key parameters to account for when modelling storage-discharge functions of hillslopes. Such characterization, although demanding, is possible through
geophysical investigation with combined surface and borehole techniques. While our approach assumes continuity of the compartment along the hillslope, another key parameter influencing stream discharge dynamics during baseflow is related to the volume and spatial connectivity of shallow and deep compartments at the catchment scale. While mechanical and bio-geochemical processes may increase the permeability and storage capacity of the bedrock, the spatial representation of this



shallow permeable compartment might be highly heterogeneous, with poor connectivity at the catchment scale (Guihéneuf et al., 2014) and, consequently, displaying higher sensitivity to droughts.

**Code and data availability**

Model results are available at Roques, C. (2022). Model results presented in HESS-2022-7, HydroShare, http://www.hydroshare.org/resource/882cbdcddfdf49ec9ba90c9e43c19cb0. Matlab codes used to process model results are available on request from the authors.

**Competing interests**

The authors declare that they have no conflict of interest.

**Author contribution**

CR, DER, GG, ERJ and JSS defined the initial research question. All co-authors were involved in conceptualization. CR developed the model, performed the simulations and analysed the results. All co-authors participated in the interpretation of
the results. CR created the figures. CR prepared the first draft of the manuscript. All co-authors reviewed and edited the manuscript.

**Acknowledgment**

Clément Roques acknowledges financial supports from the Rennes Métropole research chair "Ressource en Eau du Futur" and the European project WATERLINE, project id CHIST-ERA-19-CES-006. This work was also supported by the National
Science Foundation grant N °1551483.

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

## Appendix

### A.1 Two linear reservoirs theory:

We can define the total outflow from the drainage of 2 linear reservoirs as

$$Q(t) = Q_f(t) + Q_S(t)$$
Eq. A 1

with subscript $f$ standing for fast drainage and $s$ for slow drainage.

Linear reservoir theory defines the evolution of the discharge as a function of time:

$$Q_f(t) = Q_f(0) \exp\left(-\frac{t}{k_f}\right)$$
Eq. A 2

and





$$Qs(t) = Qs(0) \exp\left(-\frac{t}{ks}\right)$$

Eq. A 3

with $k_f$ and $k_s$ the recession constants of the fast and slow reservoirs respectively and $Q_f(0)$ and $Q_s(0)$ the initial flowrates prior recession from the fast and slow reservoirs respectively.

$k$ is related to the geometrical and hydraulic properties of the reservoir by (Boussinesq, 1877):

$$k = \frac{4\phi L^2}{\pi^2} KH$$

Eq. A 4

Considering $r_f$ the ratio of fast flow to the total flow, we can write:

$$Q(t) = Q(0)\left[\left(1 - r_f(0)\right)\exp\left(-\frac{t}{k_s}\right) + r_f(0)\exp\left(-\frac{t}{k_f}\right)\right]$$

Eq. A 5

Rearranging to solve for $r_f(t)$:yields:

$$r_f(t) = \frac{1}{1 + \left(\frac{1}{r_f(0)} - 1\right)\exp\left(\left(\frac{1}{k_f} - \frac{1}{k_s}\right)t\right)}$$

Eq. A 6

In Gao et al 2017, the authors derived the solution for solving $b$ as a function of $r_f(t)$:

$$b = \frac{1 + \left(\left(\frac{k_s}{k_f}\right)^2 - 1\right)r_f(t)}{\left(1 + \left(\frac{k_s}{k_f} - 1\right)r_f(t)\right)^2}$$

Eq. A 7

If we replace $r_f(t)$ by Eq. A6 and derive the solution for the maximum value of $b$ we find:

$$\max(b) = \frac{\left(\frac{k_s}{k_f} + 1\right)^2}{4\frac{k_s}{k_f}}$$

Eq. A 8

Considering that $k_f$ and $k_s$ are only differentiated by their hydraulic conductivities we can write **Erreur ! Source du renvoi**

**introuvable.** as a function of $r_K$.

$$\max(b) = \frac{(r_K + 1)^2}{4r_K}$$

Eq. A 9





## A.2 Sensitivity to parameter $D$ and $\alpha/\zeta$

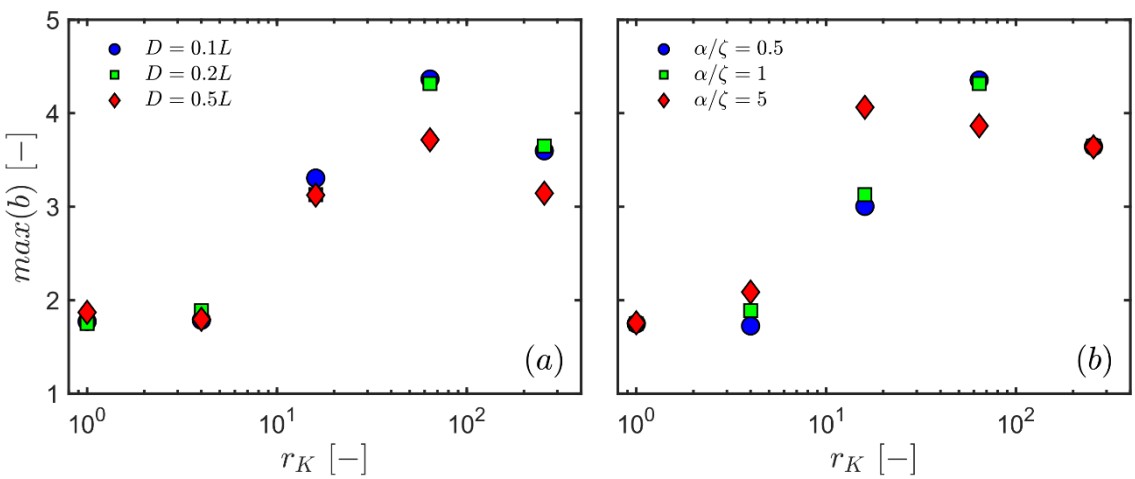

Figure 8: Maximum values of $b$ as a function of $r_K$ for $\beta = 0.2$ and (a) variations in the thickness of the upper compartment, $D$, and, (b) variations in the characteristic depth over which hydraulic properties decrease, $\alpha/\zeta$.

## A.3 Figure 7c and d displayed with a linear scale for $Q/Q_0$

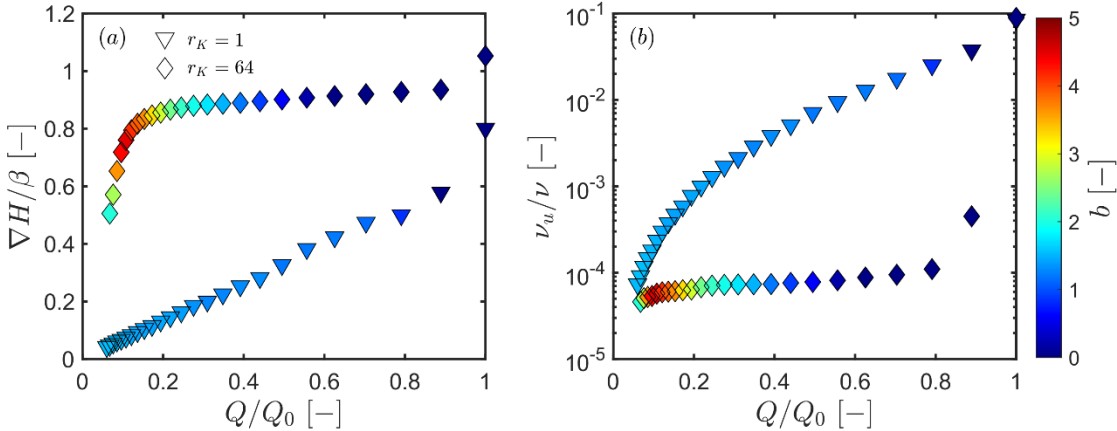

Figure 9: (a) the progression of the average head gradient $\nabla H$, normalized by the slope of the interface $\beta$, and in (b) the change in saturation area of the upper compartment $\nu_u$ with respect to the total saturation area $\nu$, as a function of $Q/Q_0$. Triangle shape symbols are for the case where $r_K = 1$ while diamond ones are for $r_K = 64$. The colours of the symbols show the magnitude of $b$
during recession.