# Peer review of "Recession discharge from compartmentalized bedrock hillslopes"

_Hydrology and Earth System Sciences, 2022_

## Referee Comment (RC1)

**Summary**

This study aims to demonstrate the importance of bedrock conductivity pattern, bedrock slope, and soil-bedrock conductivity contrast on streamflow recession using a theoretical (numerical) experiment. The authors conducted scenario analyses to explore how conductivity contrast at soil-bedrock interface, soil thickness, and porosity as well as bedrock slope impact the parameter "b" of recession analysis which shows the nonlinearity of recession response. They explored under which circumstances the homogeneous theory derived from the Boussinesq solution deems appropriate to interpret recession analysis.

**Assessment**

I enjoyed reading this manuscript. Bedrock conductivity pattern and soil-bedrock conductivity contrast could strongly control how catchments store, partition and release water and solute and thus they could control vertical distribution of flow-path and ultimately recession behavior. These controls are poorly understood and lack of consideration of such controls could lead to a misleading interpretation of recession analysis if one only relies on the homogeneous theory derived from the Boussinesq solution. This paper could be a very nice addition to HESS after some revisions and clarification.

**Major suggestion:**

This study conducted a set of analyses on the importance of parameter alpha (the rate at which Bedrock K declines exponentially) and D (upper compartment thickness or soil thickness) on recession non-linearity. But we cannot see figure or discussion on the importance of these parameters in the paper (except one fig in the appendix for max(b) only). There is one short statement in the paper that these two have had negligible impact on "*b*". I have hard time justifying this statement. Soil thickness and rate of exponential decline in K were showed to significantly control the way catchments partition and release water, particularly during low flow (see below citations). Some of the co-authors of the paper also previously showed that these two factors control recession non-linearity, using analytical solutions. I suggest that the authors further evaluate the impacts of these two and show some figures on these two parameters impact on *b*. Specifically, these two might be important at certain level of bedrock slope (beta) and soil-bedrock K contrast (rk). A sub-section of discussion can then discuss why and how the results of this paper are similar or different from previous literature. Knowing those levels of slope and rk that exaggerate the impacts of alpha and D on recession non-linearity could be very interesting.

**Suggested citations:**
Below studies focused on the importance of soil-bedrock conductivity contrast and bedrock vertical conductivity pattern (and/or soil thickness) on how catchments store, partition and release water and solute. The authors may find it helpful to use some of these citations in the introduction to further emphasize the importance of their work.

[*Ameli et al.*, 2015; *Ameli et al.*, 2016; *Ameli et al.*, 2018; *Ameli et al.*, 2021; *Ameli et al.*, 2017; *Bishop*, 1991; *Cardenas and Jiang*, 2010; *Hopp and McDonnell*, 2009; *Janssen and Ameli*, 2021; *Jiang et al.*, 2010]

Other suggestions:

Line 16: "Responsible" is a strong word. We probably cannot safely declare that yet.
Line 28-29: I don't think Tashie et al and Jachens et al suggested "*strong* dependencies".
Figure 1c. Unit is not correct

Line 119. Are results (and final conclusions) sensitive to the initial value of KL and porosity

Line 124: Identical porosity across the interface is not a conservative assumption. Could you explain/discuss how it could impact the final result and conclusions

Line 182: I think figure 4 shows as rk increases, the recessions transition from lower values of -dQ/dt to higher values. Am I missing something?

Line 290: The evidence of "compartmentalized aquifer" can lead to anomalous b values is a solid conclusion obtained from this paper. But in real world, proving that anomalous b is due to "compartmentalized aquifer" could be tricky. I suggest to revise this sentence

Line 300: But this threshold of rk was obtained given other parameters such as D and porosity and etc remained constant. So, we cannot generalize it to other situation and the threshold of rk can only be generalized for certain values of other parameters. I mean, for different D or porosity, rk might be smaller or larger than 16.

Line 305: If I am not mistaken Fig 3 conveys different message. At later time b goes toward 1 (become smaller than earlier recession). Early recession is about 1.5 and late recession is around 1.

Ameli, A. A., J. Craig, and J. McDonnell (2015), Are all runoff processes the same? Numerical experiments comparing a Darcy-Richards solver to an overland flow-based approach for subsurface storm runoff simulation, *Water Resources Research*, *51*(12).
Ameli, A. A., J. J. McDonnell , and K. Bishop (2016), The exponential decline in saturated hydraulic conductivity with depth and its effect on water flow paths and transit time distribution, *Hydrological Processes*, *30*(14), 12.
Ameli, A. A., C. P. Gabrielli, U. Morgenstern, and J. McDonnell (2018), Groundwater subsidy from headwaters to their parent water watershed: A combined field-modeling approach, *Water Resources Research*, *54*.
Ameli, A. A., H. Laudon, C. Teutschbein, and K. Bishop (2021), Where and when to collect tracer data to diagnose hillslope permeability architecture, *Water Resources Research*, *57*(8).
Ameli, A. A., K. Beven, M. Erlandsson, I. Creed, J. McDonnell, and K. Bishop (2017), Primary weathering rates, water transit times and concentration-discharge relations: A theoretical analysis for the critical zone, *Water Resources Research*, *52*.
Bishop, K. H. (1991), Episodic increases in stream acidity, catchment flow pathways and hydrograph separation, University of Cambridge.

Cardenas, M. B., and X.-W. Jiang (2010), Groundwater flow, transport, and residence times through topography-driven basins with exponentially decreasing permeability and porosity, *Water Resources Research*, *46*(11), n/a-n/a.

Hopp, L., and J. J. McDonnell (2009), Connectivity at the hillslope scale: Identifying interactions between storm size, bedrock permeability, slope angle and soil depth, *Journal of Hydrology*, *376*(3-4), 378-391.

Janssen, J., and A. A. Ameli (2021), A hydrologic functional approach for improving large-sample hydrology performance in poorly-gauged regions, *Water Resources Research*, *57*(9), e2021WR030263.

Jiang, X.-W., L. Wan, M. B. Cardenas, S. Ge, and X.-S. Wang (2010), Simultaneous rejuvenation and aging of groundwater in basins due to depth-decaying hydraulic conductivity and porosity, *Geophysical Research Letters*, *37*(5), n/a-n/a.

---

## Author Comment (AC1)

**Review report hess-2022-7.**

*Referee comments are shown with black text and the responses from the authors are in blue with italic font.*

**RC1: 'Comment on hess-2022-7', Anonymous Referee #1, 29 Mar 2022**

Summary

This study aims to demonstrate the importance of bedrock conductivity pattern, bedrock slope, and soil bedrock conductivity contrast on streamflow recession using a theoretical (numerical) experiment. The authors conducted scenario analyses to explore how conductivity contrast at soil-bedrock interface, soil thickness, and porosity as well as bedrock slope impact the parameter "b" of recession analysis which shows the nonlinearity of recession response. They explored under which circumstances the homogeneous theory derived from the Boussinesq solution deems appropriate to interpret recession analysis.

Assessment

I enjoyed reading this manuscript. Bedrock conductivity pattern and soil-bedrock conductivity contrast could strongly control how catchments store, partition and release water and solute and thus they could control vertical distribution of flow-path and ultimately recession behavior. These controls are poorly understood and lack of consideration of such controls could lead to a misleading interpretation of recession analysis if one only relies on the homogeneous theory derived from the Boussinesq solution. This paper could be a very nice addition to HESS after some revisions and clarification.

*A: We are thankful to the reviewer for the summary of the outcomes of our work and the positive assessment of the manuscript. The comments raised by the reviewer and its suggestions will be very helpful in improving our manuscript.*

Major suggestion:

This study conducted a set of analyses on the importance of parameter alpha (the rate at which Bedrock K declines exponentially) and D (upper compartment thickness or soil thickness) on recession non-linearity. But we cannot see figure or discussion on the importance of these parameters in the paper (except one fig in the appendix for max(b) only). There is one short statement in the paper that these two have had negligible impact on "b". I have hard time justifying this statement. Soil thickness and rate of exponential decline in K were showed to significantly control the way catchments partition and release water, particularly during low flow (see below citations). Some of the co-authors of the paper also previously showed that these two factors control recession non-linearity, using analytical solutions. I suggest that the authors further evaluate the impacts of these two and show some figures on these two parameters impact on b. Specifically, these two might be important at certain level of bedrock slope (beta) and soil-bedrock K contrast (rk). A sub-section of discussion can then discuss why and how the results of this paper are similar or different from previous literature. Knowing those levels of slope and rk that exaggerate the impacts of alpha and D on recession non-linearity could be very interesting.

*A: Thank you for this very important comment. We fully agree that the impact of both upper compartment thickness (D) and hydraulic conductivity decrease $K\sim z$ of the deeper compartment are important parameters in controlling flow partitioning and consequently recession behaviors. As the reviewer mentioned, Rupp and Selker [2005, 2006] have*

*previously investigated recession behaviors in aquifers with decreasing K and changing slope angles. Here, we have shown that, in the configuration of the considered in our study, the impact of upper compartment thickness and K~z of the deeper one have limited impact on max(b) with respect to the controls from r_K (figure 8, section A.2).*

*However, we agree with the reviewer that those results would deserve to be moved to the main text and further discussed. We have simulation results for 3 values of $\alpha/\zeta$ and **D** at $\beta$ = **0.2**. In the revised manuscript, we will make sure to describe those results and further develop the discussion on the impact of those parameters.*

- *To avoid confusion, we will delete the statement on Line 120 "Note that we found that changing the parameter $\alpha$ does not significantly impact our results" from the methodology section and save discussion of the influence of parameter $\alpha$ for later in the result and discussion sections.*
- *We will add a dedicated section in 3.3 where we will describe the results for $\alpha/\zeta$ and D at $\beta$ = **0.2**. We will move Figure 8 from the appendix to this section, in which we will also add a new figure showing the evolution of b vs Q for the different cases (in a similar manner as current Figure 5).*
- *We will further develop the discussion to highlight the relative importance of all parameters in controlling the variability in recession behaviors.*

Suggested citations: Below studies focused on the importance of soil-bedrock conductivity contrast and bedrock vertical conductivity pattern (and/or soil thickness) on how catchments store, partition and release water and solute. The authors may find it helpful to use some of these citations in the introduction to further emphasize the importance of their work.

[Ameli et al., 2015; Ameli et al., 2016; Ameli et al., 2018; Ameli et al., 2021; Ameli et al., 2017; Bishop, 1991; Cardenas and Jiang, 2010; Hopp and McDonnell, 2009; Janssen and Ameli, 2021; Jiang et al., 2010]

*A: Thank you for these citations. Some of them are already included in the manuscript and we will make sure to include the ones that we may have missed if appropriate.*

Other suggestions:

*A: Thank you for raising all those points. We will take them into account in our final version of the manuscript. You'll find detailed answers below when appropriate.*

Line 16: "Responsible" is a strong word. We probably cannot safely declare that yet.

Line 28-29: I don't think Tashie et al and Jachens et al suggested "strong dependencies".

Figure 1c. Unit is not correct

Line 119. Are results (and final conclusions) sensitive to the initial value of KL and porosity

*A: Thanks for raising this important point. Initial values of K_L and porosity will have limited impact as long as we stay in similar flow regimes. We will include a specific comment in the revised manuscript to highlight this important point.*

Line 124: Identical porosity across the interface is not a conservative assumption. Could you explain/discuss how it could impact the final result and conclusions

*A: Thank you for this comment. In the initial design of the study we focused on conductivity contrast. But the reviewer is right that having joined evolution between porosity and permeability will certainly have an impact. But imposing both will also come with difficulties*

*in identifying which parameter exerts most of the control. Investigating the impact of porosity contrast could be a perspective to this work, with a clear impact for solute transport mechanisms and timescales. We will add a sentence on this point.*

Line 182: I think figure 4 shows as rk increases, the recessions transition from lower values of -dQ/dt to higher values. Am I missing something?

*A: This sentence is indeed confusing – we will rephrase it.*

Line 290: The evidence of "compartmentalized aquifer" can lead to anomalous b values is a solid conclusion obtained from this paper. But in real world, proving that anomalous b is due to "compartmentalized aquifer" could be tricky. I suggest to revise this sentence

*A: We agree with this comment. Numerous factors can influence late time recession behavior and may be responsible for the emergence of anomalous behaviors. This leads to challenges when aiming to identify which factor is the most important in 'real world catchments'. However, we show, in complement to previous studies, that hillslope and catchment heterogeneities exert strong controls. We provide guidance in the discussion on key parameters/indicators that might be investigated in parallel to streamflow to identify the impact of vertical compartmentalization (lines 328-346). We will revise this sentence to highlight that this conclusion applies for the present hillslope configuration where other processes are neglected.*

Line 300: But this threshold of rk was obtained given other parameters such as D and porosity and etc remained constant. So, we cannot generalize it to other situation and the threshold of rk can only be generalized for certain values of other parameters. I mean, for different D or porosity, rk might be smaller or larger than 16.

*A: True – we will revise this sentence to highlight that it applies for the present hillslope configuration. We will add an opening sentence for perspectives to study the impact of \phi and D on changing the threshold in r_K.*

Line 305: If I am not mistaken Fig 3 conveys different message. At later time b goes toward 1 (become smaller than earlier recession). Early recession is about 1.5 and late recession is around 1.

*A: The transition from 1 to 1.5 at late times for case of \Beta = 0 is visible on figure 5a (Q<10^-6 m^2/s for r_K = 1 and 4). We will revise this sentence to specifically define the range of discharges that are concerned by those different recession behaviors.*

Ameli, A. A., J. Craig, and J. McDonnell (2015), Are all runoff processes the same? Numerical experiments comparing a Darcy-Richards solver to an overland flow-based approach for subsurface storm runoff simulation, Water Resources Research, 51(12).

Ameli, A. A., J. J. McDonnell , and K. Bishop (2016), The exponential decline in saturated hydraulic conductivity with depth and its effect on water flow paths and transit time distribution, Hydrological Processes, 30(14), 12.

Ameli, A. A., C. P. Gabrielli, U. Morgenstern, and J. McDonnell (2018), Groundwater subsidy from headwaters to their parent water watershed: A combined field-modeling approach, Water Resources Research, 54.

Ameli, A. A., H. Laudon, C. Teutschbein, and K. Bishop (2021), Where and when to collect tracer data to diagnose hillslope permeability architecture, Water Resources Research, 57(8).

Ameli, A. A., K. Beven, M. Erlandsson, I. Creed, J. McDonnell, and K. Bishop (2017), Primary weathering rates, water transit times and concentration-discharge relations: A theoretical analysis for the critical zone, Water Resources Research, 52.

Bishop, K. H. (1991), Episodic increases in stream acidity, catchment flow pathways and hydrograph separation, University of Cambridge.

Cardenas, M. B., and X.-W. Jiang (2010), Groundwater flow, transport, and residence times through topography-driven basins with exponentially decreasing permeability and porosity, Water Resources Research, 46(11), n/a-n/a.

Hopp, L., and J. J. McDonnell (2009), Connectivity at the hillslope scale: Identifying interactions between storm size, bedrock permeability, slope angle and soil depth, Journal of Hydrology, 376(3-4), 378-391.

Janssen, J., and A. A. Ameli (2021), A hydrologic functional approach for improving large-sample hydrology performance in poorly-gauged regions, Water Resources Research, 57(9), e2021WR030263.

Jiang, X.-W., L. Wan, M. B. Cardenas, S. Ge, and X.-S. Wang (2010), Simultaneous rejuvenation and aging of groundwater in basins due to depth-decaying hydraulic conductivity and porosity, Geophysical Research Letters, 37(5), n/a-n/a.

---

## Author Comment (AC2)

**Review report hess-2022-7.**

*Referee comments are shown with black text and the responses from the authors are in blue with italic font.*

**RC2: 'Comment on hess-2022-7', Anonymous Referee #2, 20 May 2022**

The paper uses numerical simulations to test the idea that a vertically compartmentalized model of groundwater flow can in some situations provide a good representation of streamflow recession, as opposed to results using the Boussinesq solution (homogeneous assumption). The paper is well written, the methodology is technically sound and the results are coherent and well presented. I only have minor comments that will hopefully help improve the paper.

*A: We are grateful to the reviewer for the very positive assessment of our work.*

In section 2.2 the authors explain that they performed CFD simulations of groundwater flow. There is no description of the model used, it is not clear whether it is the author's own model or another model. Please clarify and provide either more information or the corresponding references.

*A: We used COMSOL Multiphysics to handle and solve Richard's equations. In the revised version of the manuscript we will include more details on the model setup.*

Figure 3: I did not fully understand the explanation in the caption about the dashed lines. Maybe clearly state: solid lines correspond to…; dashed lines correspond to …

*A: We will revise the caption of Figure 3 to avoid confusions.*

Eq 4 *a* should be replaced by *a* (italic)

*A: Thank you.*

---

## Author Response (AR1)

**Review report hess-2022-7.**

*Referee comments are shown with black text and the responses from the authors are in blue with italic font.*

**Editor decision: Publish subject to minor revisions (further review by editor) by Patricia Saco**

**Comments to the author:**

We have now received the comments of two referees. I agree with the reviewers that the manuscript presents a valuable and interesting study that highlights the importance of bedrock slope and soil-bedrock conductivity on streamflow recession by using numerical experiments.

The paper is well written, and the methodology and analysis of results are well described and technically sound. There are only some minor issues that need to be addressed before considering publication, including:

1) a clearer description of sensitivity to some of the model parameters (as mentioned by reviewer #1).

2) Discussion and comparison to results from previous studies (as requested also by reviewer #1).

3) Better description of the model used in this study (referee #2)

In addition, both reviews have included some other minor comments and feedback that are very valuable. The detailed responses from the authors on how they plan to address the manuscript are very positive and show that they plan to improve the scientific contribution of the manuscript following the referees comments.

*Dear Editor,*

*We are thankful for the very positive assessment of our manuscript. The review process enabled very valuable comments that have helped us improving the manuscript. In this new version of the manuscript, we have carefully addressed all comments. We put emphasis to answer reviewer#1 comments (points 1,2, see below) and precise the model used in the study (point 3)*

*Sincerely,*

*Clément Roques, on behalf of all co-authors.*

**RC1: 'Comment on hess-2022-7', Anonymous Referee #1, 29 Mar 2022**

Summary

This study aims to demonstrate the importance of bedrock conductivity pattern, bedrock slope, and soil bedrock conductivity contrast on streamflow recession using a theoretical (numerical) experiment. The authors conducted scenario analyses to explore how conductivity contrast at soil-bedrock interface, soil thickness, and porosity as well as bedrock slope impact the parameter "b" of recession analysis which shows the nonlinearity of recession response. They explored under which circumstances the homogeneous

theory derived from the Boussinesq solution deems appropriate to interpret recession analysis.

Assessment

I enjoyed reading this manuscript. Bedrock conductivity pattern and soil-bedrock conductivity contrast could strongly control how catchments store, partition and release water and solute and thus they could control vertical distribution of flow-path and ultimately recession behavior. These controls are poorly understood and lack of consideration of such controls could lead to a misleading interpretation of recession analysis if one only relies on the homogeneous theory derived from the Boussinesq solution. This paper could be a very nice addition to HESS after some revisions and clarification.

*A: We are thankful to the reviewer for the summary of the outcomes of our work and the positive assessment of the manuscript. The comments and suggestions raised by the reviewer were very helpful in improving the message of our manuscript.*

Major suggestion:

This study conducted a set of analyses on the importance of parameter alpha (the rate at which Bedrock K declines exponentially) and D (upper compartment thickness or soil thickness) on recession non-linearity. But we cannot see figure or discussion on the importance of these parameters in the paper (except one fig in the appendix for max(b) only). There is one short statement in the paper that these two have had negligible impact on "b". I have hard time justifying this statement. Soil thickness and rate of exponential decline in K were showed to significantly control the way catchments partition and release water, particularly during low flow (see below citations). Some of the co-authors of the paper also previously showed that these two factors control recession non-linearity, using analytical solutions. I suggest that the authors further evaluate the impacts of these two and show some figures on these two parameters impact on b. Specifically, these two might be important at certain level of bedrock slope (beta) and soil-bedrock K contrast (rk). A sub-section of discussion can then discuss why and how the results of this paper are similar or different from previous literature. Knowing those levels of slope and rk that exaggerate the impacts of alpha and D on recession non-linearity could be very interesting.

*A: Thank you for this very important comment. We fully agree that the impact of both upper compartment thickness (D) and hydraulic conductivity decrease K~z of the deeper compartment are important parameters in controlling flow partitioning and consequently recession behaviors. As the reviewer mentioned, Rupp and Selker [2005, 2006] have previously investigated recession behaviors in aquifers with decreasing K and changing slope angles. Here, we have shown that, in the configuration of the considered in our study, the impact of upper compartment thickness and K~z of the deeper one have limited impact on max(b) with respect to the controls from r_K (figure 8, section A.2).*

*However, we agree with the reviewer that those results deserve to be moved to the main text and further discussed. We have now included simulation results for 3 values of $\alpha/\zeta$ and **D** at $\beta$ = 0.2 in the main text. In the revised manuscript, we have made the following changes:*

- *To avoid confusion, we have deleted the statement on Line 120, "Note that we found that changing the parameter $\alpha$ does not significantly impact our results", from the methodology section and save discussion of the influence of parameter for later in the result and discussion sections.*

- *We have added a dedicated section in 3.3 where we describe the results for $\alpha/\zeta$ and D at $\beta = 0.2$ with a new figure showing the evolution of b vs Q.*

Suggested citations: Below studies focused on the importance of soil-bedrock conductivity contrast and bedrock vertical conductivity pattern (and/or soil thickness) on how catchments store, partition and release water and solute. The authors may find it helpful to use some of these citations in the introduction to further emphasize the importance of their work.

[Ameli et al., 2015; Ameli et al., 2016; Ameli et al., 2018; Ameli et al., 2021; Ameli et al., 2017; Bishop, 1991; Cardenas and Jiang, 2010; Hopp and McDonnell, 2009; Janssen and Ameli, 2021; Jiang et al., 2010]

*A: Thank you for these citations. The manuscript already included Cardenas and Jiang (2010) and Ameli et al. (2016). In the new version we added Ameli et al., (2021) and Hopp and McDonnell (2009).*

Other suggestions:

*A: Thank you for raising all those points. We will take them into account in our final version of the manuscript. You'll find detailed answers below.*

Line 16: "Responsible" is a strong word. We probably cannot safely declare that yet.

*A: changed by "controlling".*

Line 28-29: I don't think Tashie et al and Jachens et al suggested "strong dependencies".

*A: we have changed this sentence that now use the same terminology as in Tashie et al 2020 and Jachens et al. 2020:*

*"They have also shown that such non-linearities in storage-discharge functions may be variable across different recharge events, suggesting dependencies on antecedent catchment conditions (Jachens et al., 2020; Tashie et al., 2020)."*

Figure 1c. Unit is not correct

*Modified*

Line 119. Are results (and final conclusions) sensitive to the initial value of KL and porosity

*A: Thanks for raising this important point. Initial values of K_L and porosity will have limited impact as long as similar flow regimes are involved. In the revised manuscript, we have added the following sentence in the discussion.*

*"With $K_L$ chosen at $5\ 10^{-6}\ m/s$, simulated recession behaviours are typical of fractured igneous and metamorphic bedrocks (Dewandel et al., 2006). Initial values of $K_L$, as well as the porosity $\theta_L$, may impact results and interpretation if different flow regimes are involved when increasing $r_K$. Exploring different values of $K_L$ and $\theta_L$ typical of other lithologies is also an important perspective of this work."*

Line 124: Identical porosity across the interface is not a conservative assumption. Could you explain/discuss how it could impact the final result and conclusions

*A: Thank you for this comment. The reviewer is right that having a fixed porosity for the upper compartment is not a conservative assumption. Considering joined evolution*

*between porosity and permeability is expected to have an impact on recession behavior but will also come with difficulties when identifying which parameter exerts most of the control. This is why we decided to focus on changing only K_U. Investigating the impact of porosity contrast could be a perspective to this work, with a clear impact for solute transport mechanisms and timescales.*

*We added the following sentence in discussion:*

*"Similarly, our experiment considers a constant value of porosity for the upper compartment across simulations. While this assumption neglects the coevolution of $\theta_U$ and $K_U$ during geomechanically and geochemically-driven processes shaping the upper compartment of the aquifer, it allowed us to focus our interpretation on the impact of hydraulic conductivity enhancement. Investigating the impact of porosity contrast remains to be explored."*

Line 182: I think figure 4 shows as rk increases, the recessions transition from lower values of -dQ/dt to higher values. Am I missing something?

*A: Indeed – we have modified the sentence as follow:*

*"Considering cases where $\beta > 0$ and by increasing $r_K$, values of $-dQ/dt$ involved at the onset of the recessions increases."*

Line 290: The evidence of "compartmentalized aquifer" can lead to anomalous b values is a solid conclusion obtained from this paper. But in real world, proving that anomalous b is due to "compartmentalized aquifer" could be tricky. I suggest to revise this sentence

*A: This paragraph aims at discussing the impact of bedrock heterogeneity. We agree that there are challenges when aiming to identify which factor (climatic, geomorphic, geologic) dominates in "real world catchments". We actually discuss this point in the last paragraph of the discussion. In complement, the manuscript also provides guidance on key indicators that might be investigated in parallel to streamflow recession to identify the controls form vertical compartmentalization (lines 328-346 of initial submission).*

Line 300: But this threshold of rk was obtained given other parameters such as D and porosity and etc remained constant. So, we cannot generalize it to other situation and the threshold of rk can only be generalized for certain values of other parameters. I mean, for different D or porosity, rk might be smaller or larger than 16.

*A: We have modified this sentence. It reads now:*

*"The assumption is satisfied when lateral flow in the shallower compartment prevails, i.e. with only limited contribution from the deep bedrock to the discharge dynamics (when $r_K \geq 16$ in our hillslope configuration)."*

Line 305: If I am not mistaken Fig 3 conveys different message. At later time b goes toward 1 (become smaller than earlier recession). Early recession is about 1.5 and late recession is around 1.

*A: The transition from 1 to 1.5 at late times for case of \Beta = 0 is visible on figure 5a (Q<10^-6 m^2/s for r_K = 1 and 4). We completed the sentence as follow:*

*"At later times, $b$ values were found to increase again controlled by the decay in hydraulic conductivity as shown in Figure 5a for $Q < 10^{-6} m^2/s$ and $r_K = 1$ and 4."*

Ameli, A. A., J. Craig, and J. McDonnell (2015), Are all runoff processes the same? Numerical experiments comparing a Darcy-Richards solver to an overland flow-based approach for subsurface storm runoff simulation, Water Resources Research, 51(12).

Ameli, A. A., J. J. McDonnell , and K. Bishop (2016), The exponential decline in saturated hydraulic conductivity with depth and its effect on water flow paths and transit time distribution, Hydrological Processes, 30(14), 12.

Ameli, A. A., C. P. Gabrielli, U. Morgenstern, and J. McDonnell (2018), Groundwater subsidy from headwaters to their parent water watershed: A combined field-modeling approach, Water Resources Research, 54.

Ameli, A. A., H. Laudon, C. Teutschbein, and K. Bishop (2021), Where and when to collect tracer data to diagnose hillslope permeability architecture, Water Resources Research, 57(8).

Ameli, A. A., K. Beven, M. Erlandsson, I. Creed, J. McDonnell, and K. Bishop (2017), Primary weathering rates, water transit times and concentration-discharge relations: A theoretical analysis for the critical zone, Water Resources Research, 52.

Bishop, K. H. (1991), Episodic increases in stream acidity, catchment flow pathways and hydrograph separation, University of Cambridge.

Cardenas, M. B., and X.-W. Jiang (2010), Groundwater flow, transport, and residence times through topography-driven basins with exponentially decreasing permeability and porosity, Water Resources Research, 46(11), n/a-n/a.

Hopp, L., and J. J. McDonnell (2009), Connectivity at the hillslope scale: Identifying interactions between storm size, bedrock permeability, slope angle and soil depth, Journal of Hydrology, 376(3-4), 378-391.

Janssen, J., and A. A. Ameli (2021), A hydrologic functional approach for improving large-sample hydrology performance in poorly-gauged regions, Water Resources Research, 57(9), e2021WR030263.

Jiang, X.-W., L. Wan, M. B. Cardenas, S. Ge, and X.-S. Wang (2010), Simultaneous rejuvenation and aging of groundwater in basins due to depth-decaying hydraulic conductivity and porosity, Geophysical Research Letters, 37(5), n/a-n/a.

**RC2: 'Comment on hess-2022-7', Anonymous Referee #2, 20 May 2022**

The paper uses numerical simulations to test the idea that a vertically compartmentalized model of groundwater flow can in some situations provide a good representation of streamflow recession, as opposed to results using the Boussinesq solution (homogeneous assumption). The paper is well written, the methodology is technically sound and the results are coherent and well presented. I only have minor comments that will hopefully help improve the paper.

*A: We are grateful to the reviewer for the very positive assessment of our work.*

In section 2.2 the authors explain that they performed CFD simulations of groundwater flow. There is no description of the model used, it is not clear whether it is the author's own model or another model. Please clarify and provide either more information or the corresponding references.

*A: In the revised manuscript we have included that we used COMSOL Multiphysics to handle and solve Richard's equations. It reads as:*

*We performed Computational Fluid Dynamics (CFD) simulations in the 2-dimensional (2D) hillslope. The flow for variably saturated porous media with incompressible fluid is modelled by solving the Richards' equations with a finite element approach implemented in COMSOL Multiphysics®.*

Figure 3: I did not fully understand the explanation in the caption about the dashed lines. Maybe clearly state: solid lines correspond to…; dashed lines correspond to …

*A: We have now modified the caption of Figure 3. It reads as follow:*

*"Recession plots displaying $-dQ/dt$ vs $Q$ in a log-log space obtained for the horizontal interface, i.e. $\beta = 0$, and different hydraulic conductivity ratios $r_K$. The dashed lines show predictions for a flat and homogeneous aquifer without the deep compartment combining equations (1) and (4) ($b = 1.5$; Boussinesq 1904). The slope for $b = 1$, characteristic of the drainage of a linear reservoir, has been drawn for reference."*

Eq 4 a should be replaced by *a* (italic)

*A: Modified*